# Food Insecurity in the Informal Settlements of Inanda Households Living with Children under 60 Months in Ethekwini Municipality

**DOI:** 10.3390/children9101521

**Published:** 2022-10-05

**Authors:** Mbalenhle Mkhize, Melusi Sibanda

**Affiliations:** Department of Agriculture, University of Zululand, Private Bag X1001, KwaDlangezwa 3886, South Africa

**Keywords:** children under five years, food insecurity, informal settlements, South Africa, poor urban settlements

## Abstract

Food insecurity is a continuing challenge for many households in South Africa. This challenge poses serious immediate and long-term health and development risks for children. Despite the intensive literature on household food insecurity, there is limited literature on the household food security status in South African informal settlements. Thus, the household food security status and dynamics in informal settlements are not clearly defined. Hence, this study assessed the food security status of households living with children under 60 months in the informal settlements of the Inanda area, eThekwini Municipality. This study employed a cross-sectional quantitative research approach. A non-probability sampling method was used, which used convenience sampling supplemented by a non-discriminative snowball sampling to obtain a sample size of 160 households with children under the age of five. Data was collected through face-to-face interviews, where questionnaires were administered to household child caregivers. Ethical considerations such as informed consent, anonymity, confidentiality, permission from authorities, and cultural considerations were obeyed in this study. The HFIAS and HDDS tools were used to estimate the household food security status. Data were coded and analysed in SPSS version 25 software. This study revealed that higher proportions of the surveyed informal households living with children under 60 months were food insecure. The HFIAS analysis showed that approximately 34, 31, and 28% were severely, mildly, and moderately food insecure, respectively. In contrast, a small (approximately 8%) proportion of the surveyed informal households was estimated to be food secure. The HDDS analysis revealed that most (approximately 77%) of the surveyed informal households had low dietary diversity (deemed food insecure). Cereal, roots, and fatty foods were the main dietary components in the informal settlements of Inanda. It is paramount to improve the food security status of informal households living with children under 60 months through an integrated approach. This study suggests government and private stakeholders’ engagement in developing policies and programs directed at informal households living with children under 60 months to alleviate food insecurity.

## 1. Introduction

For the past decades, household food insecurity has gained much attention globally. However, household food insecurity is an alarming burden for many African countries, including South Africa [1]. Recent reports show an increase in food insecurity, whereby 2.3 billion people globally were severely food insecure in 2021 [2]. The rise in the number of food insecure people evidences the failure to achieve the goal of ending hunger, food insecurity, and malnutrition by 2030. Although South Africa is food secure at the national level, the country is still food insecure at the household level [3]. In 2020, South Africa had more than one-half million households with children under five years who were food insecure [4]. The increase in household food insecurity in South Africa is associated with various factors, such as the COVID-19 pandemic, economic downfall, and climate change [5]. Recent scholars have identified a shift in food insecurity in rural areas to poor urban settings [6,7,8]. This shift may be due to increased households residing in informal settlements. According to Abebe et al. [9], the growth of informal settlements is synonymous with urbanization. In South Africa, approximately 64 percent of households live in informal settlements [10]. Informal settlements are socially and economically marginalized places with high poverty levels and poor sanitation [11]. According to Ngcamu and Mantzaris [12], households in informal settlements were least prepared for the pandemic of COVID-19 because of the short supply or non-existence of basic needs such as water and sanitation. Corburn and Sverdlik [13] explained that households in informal settlements are excluded from urban development and opportunities, making it hard to break the cycle of poverty and food security. Therefore, the household food security status in the South African informal settlement is not clearly defined. As a result, there are limited policy interventions that address food insecurity in informal settlements.

Food insecurity has negative impacts on children, especially during their developmental ages. These can be immediate or long-term [14]. The consequences may range from physical and mental disorders, poor academic performance, attention problems, absenteeism from school, and other behavioural challenges [15]. Although there is much research and literature on food insecurity, less focus has been paid to understanding food insecurity and its dynamics for households living with children. Food security is significant for children because nutrition greatly influences their health and future well-being [16]. Guided by the literature review, this study hypothesized that households in informal settlements of the Inanda area, eThekwini Municipality, are food insecure. Understanding the food insecurity status of households with children and the associated socioeconomic dynamics is critical. Such an understanding contributes to framing effective intervention programs that will support the well-being of children in vulnerable informal settlement households. Hence, this study aims to fill this gap by assessing household food security status in the informal settlements of the Inanda area, eThekwini Municipality. 

Generally, food security refers to reliable access to adequate, healthy, and culturally appropriate food. While on the other extreme, food insecurity is the opposite, lacking or inability to obtain good, healthy, and culturally acceptable food [17]. In other words, food insecurity is at the lower spectrum (low to very low) of the food security status. This study adopted the household food security conceptual framework that Bashir and Schilizzi [18] developed. This conceptual framework acts as a guideline to determine the food security status at a local level. It is mainly based on three broad food security interlinked components: food availability, accessibility, and utilization. Food availability is referred to as the physical presence of food [19]. Food availability includes the self-production of food and market purchases [20]. Self-production of food is determined by the availability of natural resources, inputs, and credit to practice agricultural activities [21]. Meanwhile, market purchases depend on the market dynamics to obtain food [22].

## 2. Materials and Methods

The study was carried out in the informal settlements of the Inanda area: Bhambayi, Congo 1, Dikiwe, Dube Village, Ekuphakameni, Mission, and Simunye Triangle. The Inanda area was purposefully selected because of its poor socioeconomic status, low agricultural potential, and overpopulation. Inanda is classified as one of the largest low-income residential areas in South Africa [23]. Thirty percent (30%) of individuals are employed, 42 percent are economically inactive, and 28 percent are unemployed [24]. Findings of Tshishonga [25] show that many individuals in Inanda are illiterate and unskilled, making it impossible to break the cycle of poverty. Figure 1 shows the study area of the Inanda informal settlements in eThekwini Municipality. 

### 2.1. Conceptual Framework

Figure 2 shows the conceptual framework of household food security.

According to Ristaino et al. [27], global food availability is stabilized above adequate levels; however, food insecurity is persistent. This situation implies that food availability is vital, but access to food by individuals is the primary constraint. According to Holbel and Marshalle [28], accessibility of food is ensured when all household members have adequate resources to obtain sufficient nutritious foods. The determinants of food accessibility at the household level include economic status, demographic characteristics, and food prices. The economic elements consist of income distribution within the household, employment status, sources of income, and purchasing power. The household demographic characteristics include the educational levels of a household head, gender, household sizes and the total number of dependents [29]. Ultimately, household food security status is influenced by food utilization. Food utilization refers to an individual’s dietary intake and capacity to absorb nutrients contained in the food consumed [30]. Food utilization includes the quantity and quality of food. Therefore, food utilization covers an individual’s dietary intake, safety, and health status [31].

### 2.2. Sampling and Data Collection

This study employed a cross-sectional quantitative research approach. A sample size of 160 households with children 24–60 months old was used. This sample size was deemed acceptable considering similar studies which estimated household food insecurity; for example, [32] sampled 160 households, and the findings were significant. This sample size was obtained using a non-probability sampling procedure, where a convenience sampling method supplemented by a snowball sampling method was used. Data were collected through face-to-face interviews, where questionnaires were administered to household child caregivers. The response rate of questionnaires was 100 percent. Questionnaires were pre-tested before the commencement of data collection. Pre-testing of questionnaires was conducted from 16 December 2019 to 29 December 2019. Pre-testing was carried out on 20 households with children under 5 years. Before data collection, permission (ethical clearance) from the University of Zululand was obtained. Approval was also granted by the KZN Department of Social Development (DSD).

In addition, permission was also given by local authorities. Data were collected for 7 weeks, from 6 January 2020 to 28 February 2020. After data collection, raw data was captured and encoded in a spreadsheet in Microsoft excel. For analysis, data were exported to Statistical Package for Social Science (SPSS) version 25. Household Food Insecurity Access Scale (HFIAS) and Household Dietary Diversity Score (HDDS) were used to estimate the household food insecurity status in households situated in informal settlements of Inanda. A Chi-Square test and the Pearson correlation between household food security status (HFIAS) and selected explanatory variables (household and respondents’ demographic characteristics) were used for further analysis. Pearson’s correlation coefficients with *p* < 0.05 were considered significant. 

### 2.3. Food insecurity Estimation

#### 2.3.1. Household Food Insecurity Access Scale (HFIAS)

The HFIAS is a simple and accurate tool for measuring a household’s accessibility to adequate food [33]. The HFIAS consists of 9 linked occurrence questions that determine the prevalence of household food insecurity. The 9 occurrence questions can be used to reveal different food insecurity experiences, such as: 

Having mixed emotions and apprehension over food;

Observing that the available food is insufficient for both grown-ups and youngsters;

Noticing that food does not contain the dietary diversity that is required;

There has been a substantial decrease in the consumption of food;

Stated concerns about a reduction in food consumption of both adults and children and

Thinking of malicious or improper methods to acquire food for consumption [34]. 

According to Coates et al. [35], four indicators that are used to assist in understanding the features and variations of food insecurity are as follows: 

Access scale score;

Access related conditions;

Access prevalence and

Access-related domains. 

The HFIAS score represents the degree or level at which a household is food insecure during a 30-day period. The household HFIAS score is calculated by summing the frequency of occurrence codes for each question. The HFIAS score ranges from 0 to 27. To determine the food insecurity status, households with the lowest average HFIAS score of 0–1 are considered food secure. Households with an average HFIAS score of 2 to 8 are considered food insecure mildly. A household with an average HFIAS score from 9 to 16 is considered moderately food insecure, and an average HFIAS score from 17 to 27 is severely food insecure. The following equation shows how the HFIAS was estimated (Equation (1)):(1)HFIAS Score=Sum frequency of occurence question response codes (Q1a+Q2a +Q3a+Q4a+Q5a+Q6a+Q7a+Q8a+Q9a)
where *HFIAS score* = sum of the occurrence frequency during the past 30 days for the 9 food insecurity-related conditions. The same applies to *Q1a*, *Q2a*, *Q3a*, *Q4a*, *Q5a*, *Q6a*, *Q7a*, *Q8a*, and *Q9a*. *Q1a* to *Q9a* denote the nine occurrence questions from the HFIAS tool. The *average HFIAS* is, therefore, calculated as follows (Equation (2)):(2)$$Avarage\;HFIAS\;score=\frac{Sum\;of\;HFIAS\;scores\;in\;the\;sample}{\;Number\;of\;HFIAS\;scores\;\left(i\;e.\;household\right)\;in\;the\;sample}$$

#### 2.3.2. Household Dietary Diversity Score (HDDS)

Household food insecurity estimation was also achieved by using the HDDS tool. The HFIAS estimation alone does not refer directly to the household’s nutrient uptake quality. In contrast, the HDDS estimation indicates the household’s access to various food and nutrients while determining the food security status. The HDDS is considered a qualitative measure of different food groups consumed over a particular period and reflects household access to various foods [36]. The HDDS questionnaire reflects a rapid, user-friendly, and easily administered low-cost assessment tool [37]. 

Data for the HDDS indicator were collected by questioning the respondent with a series of “Yes” or “No” questions where “Yes” was coded a numeric value of 1 and “No” was coded as 0. According to Swindale and Bilinsky [38], HDDS can measure a household’s socioeconomic level by scoring based on 12 food groups. To calculate the HDDS, the following set of 12 food groups were used: A = Cereal, B = Root and tubers, C = Vegetables, D = Fruits, E = Meat, poultry, and offal, F = Eggs, G = Fish and seafood, H = Pulses, legumes, and nuts, I = Milk and dairy product, J = Oil and fats, K = Sugar and honey, and L = Miscellaneous. The number of food groups consumed in the household over a 24-h recall period was summed to estimate the HDDS. The HDDS variable was calculated for each household. The value of the HDDS variable ranged from 0 to 12. For the total number of food groups consumed by the household, food groups were represented by letters from A through L. Respondents were asked to indicate a “Yes” if they had consumed that particular food group over 24 h. The respondents indicated a “No” if they had not consumed a specific food group over the same recall period. The sum of the HDDS is calculated as follows (Equation (3)): (3)HDDS=A + B + C + D + E + F + G + H + I + J + K + L

The average HDDS was then estimated as follows (Equation (4)):(4)$$\mathrm{Average}\;\mathrm{HDDS}=\frac{\mathrm{Sum}\;\left(\mathrm{HDDS}\right)}{\mathrm{Total}\;\mathrm{number}\;\mathrm{of}\;\mathrm{households}}$$

If income records from the survey are not accessible, an HDDS target can be obtained by taking the average HDDS of the household [39]. For this study, household income levels were inaccessible, and the HDDS target was attained from the average HDDS.

#### 2.3.3. Limitations of the Study

The study was not immune to challenges, which hindered it from reaching its full potential. Respondents had a fear of participation in the study. For one, some feared that the researcher was an undercover investigator authorised by the municipality. Other respondents believed that the researcher was a government stakeholder who came to make empty promises. The researcher, however, emphasised that she was a student studying factors contributing to child malnutrition and that any information obtained would only be used for research purposes. 

## 3. Results 

### 3.1. Households with Children and Respondents’ Demographic Characteristics 

Findings revealed that a greater (31.30%) share of respondents in households with children were between the ages of 31 and 45. Approximately one-quarter (25.60%) of respondents were between the ages of 26 and 30 (Table 1). There were 18.10, 12.50, and 10.00 percent of respondents (in households with children) aged 19–25, 46–50, and over 50 years, respectively. A small proportion (2.50%) of respondents who were in households with children were less than 19 years of age. Almost half (43.10%) of respondents in households with children had a primary school level of education (Table 1). Approximately thirty-eight (37.50) percent of respondents in households with children never went to school. Fifteen (15) percent of respondents (caregivers) in households with children had a secondary level of education, and 4.40 percent of respondents had a non-formal education (i.e., a designed learning situation that does not have a curriculum, syllabus, accreditation, or certification but enhances skill development such as community craft programs) (Table 1). 

Results of this study show that a high proportion (69.40%) of respondents in households with children were unemployed. There were 13.10 percent of self-employed and temporarily employed respondents, with only a few (4.40%) of the respondents permanently employed (Table 1). Meanwhile, a high (95.60%) percentage of respondents in households with children received a social grant. Only a few (4.40%) respondents did not receive any form of social grant assistance. Finally, slightly above half (52.5%) of respondents in households with children had a total number of dependents between 1 and 5 members, and 47.50 percent of respondents had between 6 and 10 members as dependents (Table 1).

### 3.2. The Food Security Status of Households with Children in the Inanda Informal Settlements

#### Results of the Household Food Insecurity Access Scale Analysis

The findings of this study show that a high (34.40%) percentage of households with children were severely food insecure, 30.60 percent were mildly food insecure, and 27.50 percent were moderately food insecure (Table 2). A low (7.50%) percentage of respondents’ households with children were food secure. The mean score for the HFIAS was 11.80, with a standard deviation of 7.52 (Table 2). Table 2: The food insecurity access scale of households with children in the Inanda informal settlement.

### 3.3. Results of the Household Dietary Diversity Score Analysis

Findings from the HDDS analysis show that approximately 99.38 percent of respondents in households with children indicated that the households consumed cereals, and 96.86 percent of respondents had consumed oil or fats (Table 3). An estimated 89.38, 73.75, 68.75, and 61.88 percent of respondents in households with children indicated that the households consumed roots or tubers, miscellaneous, and sugar or honey, respectively (Table 3). Furthermore, 55.63, 38.15, 20.63, 25, 13.13, and 13.13 percent of respondents in households with children indicated that the households consumed pulses or legumes, fruits, meat or poultry, eggs, milk, and fish or seafood, respectively (Table 3). Table 3 shows the food groups consumed by households with children in the Inanda informal settlements during a 24-h recall period. The HDDS was categorized into low, medium, and high dietary diversity. A low dietary diversity household consumed less than four food groups. A medium dietary diversity household consumed not more than six food groups, and a high dietary diversity household consumed more than six food groups. Households that fell into low dietary diversity were deemed to be food insecure. Households that fell into medium dietary diversity were considered moderately food secure, and households that fell into high dietary diversity were classified as food secure. Approximately 76.90 percent of the households with children fell into the low dietary diversity, 11.90 percent fell into the medium dietary diversity, and 11.30 fell into the high dietary diversity (food secure) (Table 4). The average HDDS was 5.09, with a standard deviation of 1.28 (Table 4).

### 3.4. Chi-Square Test and Pearson Correlation between Household Food Security Status (HFIAS) and Selected Explanatory Variables

The Chi-Square test and Pearson correlation showed a significant relationship between the caregiver’s age in households with children and food security status at *p* = 61.878 and *r* = 0.000. Caregivers aged 31–45 in households with children had the highest severe food insecurity percentage (11.88%) (Table 5). Caregiver’s level of education in households with children was significant at *p* = 9.926 and *r* = 0.019 (Table 5). A high (18.13%) percent of caregivers who never went to school in households with children were severely food insecure (Table 5). The caregivers’ employment status in households with children was significant at *p* = 42.09 and *r* = 0.000. Unemployed caregivers in households with children were more (23.7%) severely food insecure (Table 5). Caregivers’ access to social grants in households with children was also significant at *p* = 7.028 and *r* = 0.008. A higher percentage (34.38%) of caregivers who received social grants in households with children were severely food insecure (Table 5). The total household number of dependents in households with children was insignificant at *p* = 1.679 and r = 0.195 (Table 5). Nonetheless, there was a high level (20.63%) of severe food insecurity among caregivers in households with children, with a total household number of dependents from one to five (Table 5).

## 4. Discussion

The findings of this study revealed that most respondents in households with children were aged 31–45 years. This finding suggests that most respondents in households with children are still in their productive economic cycle. They stand a chance to be economically active and improve their household food security. The Chi-Square and Pearson correlation analyses showed that respondents’ age in households with children was significantly associated with household food security status. Mahlangu and Chelule [40] also noted a significant association between caregivers’ age and household food security status. In contrast, Sekhampu [41] did not establish any association between age and household food security status. There seems to be no consensus among scholars on the association between age and household food security status.

Findings of this study showed that more respondents in households with children had a primary level of education, followed by those who never went to school. The Chi-Square analysis showed a significant association between respondents’ level of education in households with children and household food security status, in which most of the respondents were illiterate and severely food insecure. This study’s findings align with Hoq et al. [42], who demonstrated that one in every three caregivers with food-insecure households is illiterate. Another study in China showed a significant association between caregivers’ illiteracy and household food security [43]. Meanwhile, a survey in Northeast Ethiopia illustrated that households with illiterate caregivers suffered from food insecurity [44]. Illiterate caregivers in households with children are more likely to be unskilled and unemployed and, thus, contribute less to household income [45]. Similarly, Swanepoel et al. [46] explained that caregivers’ illiteracy is closely associated with low household income, influencing household food security. 

This study established a significant association between the caregiver’s access to social grants in households with children and household food security status. This study showed a significantly high number of respondents who had access to social grants in households with children to be severely food insecure. This discovery demonstrated that although most respondents in households with children depended on social grants as their source of income, they were still prone to food insecurity. Global scholars also show food insecurity amongst households relying on social grants as a source of income [47,48]. Although social grants in South Africa have at least improved the poverty status, food insecurity incidents continue to be a massive challenge for many South Africans [49]. 

The high number of household dependents influences the household’s food security status as the family requires a higher income to support the members. In this study, households with children with a total number of dependents equal to or above six (6) were deemed large, while those with a total number of dependents less than six were considered small. The results of this study illustrated a substantial number of respondents in households with children with few household dependents. However, there was an insignificant association between the total number of dependents in households with children and household food security status. The study’s findings diverge from similar studies, for example, Kalu and Etim [50] and Kaoje et al. [51]. Again, Galgamuwa et al. [52] and Park et al. [53] revealed that many economically inactive household members are a risk for household food insecurity. 

This study revealed food insecurity in informal settlement households with children under five in the Inanda area. The HFIAS analysis showed that only a few households with children were food secure while the rest of the study population was food insecure. A significant proportion was severely food insecure. Although there were a few studies conducted in South African informal settlements on household food insecurity, results from similar studies conducted in South African informal settlements—such as Crush and Caesar [54], Naicker et al. [55], and Hunter-Adams [56]—align with the results of this study. A high prevalence of household food insecurity is noted in informal settlements. Food insecurity may be due to poor socioeconomic status and service delivery in informal settlements. 

Household dietary diversity scores have been validated to be reliable in describing dietary intake and household food insecurity [57]. Findings from this study showed that households with children in the informal settlements of Inanda lack diversity in their diet. The main foods consumed are cereals, roots or tubers, and foods made with oil or fats. This finding suggests that their diet mainly consisted of foods highly concentrated in carbohydrates, starches, and fats. The consumption of carbohydrate-rich foods, starch, and fats could be due to local cultural practices. Results showed a low consumption of dairy products such as milk. There was also a very low consumption of foods such as eggs, seafood, meat, and fruits. This situation may be due to the inability to purchase these foods deemed as costly in these situations. This finding demonstrates that households with children in informal settlements of Inanda have poor access to sufficient nutritious food. This discovery classifies households in informal settlements of Inanda as food insecure. A similar study by Rakotonirainy et al. [58] reported similar results showing that most food insecure households had poorly diversified diets consisting of foods high in carbohydrates and deprived of meat products. Ochieng [59] also showed similar results. These findings support the argument that households that lack dietary diversity are food insecure and at risk of poor health. Therefore, a diverse diet is vital for improving household food security status and, thus, health status.

## 5. Conclusions and Recommendations 

The findings of this study illustrate high illiteracy and unemployment, where most respondents received social grants in households with children. A large number of dependents was also observed in households with children. This finding implies that households with children in the Inanda area informal settlements may have a low household income at their disposal, making them vulnerable to household food insecurity. The results of this study revealed household food insecurity in the informal settlement in households living with children in the Inanda area. This observation is explained by the high number of severely food insecure households. Low dietary diversity was prevalent amongst most households, where mainly carbohydrates, starch, and fats as the main foods were consumed. The findings generally show that most sampled informal households living with children were food insecure regarding access and dietary diversity. This finding implies that children in the food-insecure vulnerable households of the informal settlements are likely to face elevated risks of health and development problems. Based on these findings, the social-economic situation of households in informal settlements should improve job creation and skills development. This study also suggests developing and enacting improved policies and programs to enhance household food security. This intervention can be achieved by improving livelihoods, focusing on agricultural production and improving the resilience of marginalized populations. Again, we recommend household food and nutrition assistance programs to improve children’s food security by providing informal settlement households with children under age five access to healthy food and nutrition education. The direction for future research would be to assess household food security status for the marginalized areas, particularly informal settlements incorporating macro- and micro-factors not covered in this study.

## Figures and Tables

**Figure 1 children-09-01521-f001:**
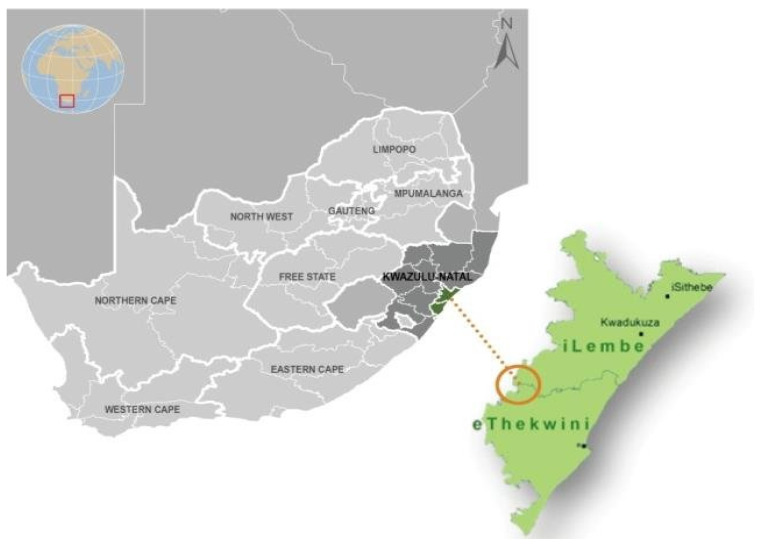
Map showing the location of Inanda to eThekwini Municipality. Source: Lincoln [26].

**Figure 2 children-09-01521-f002:**
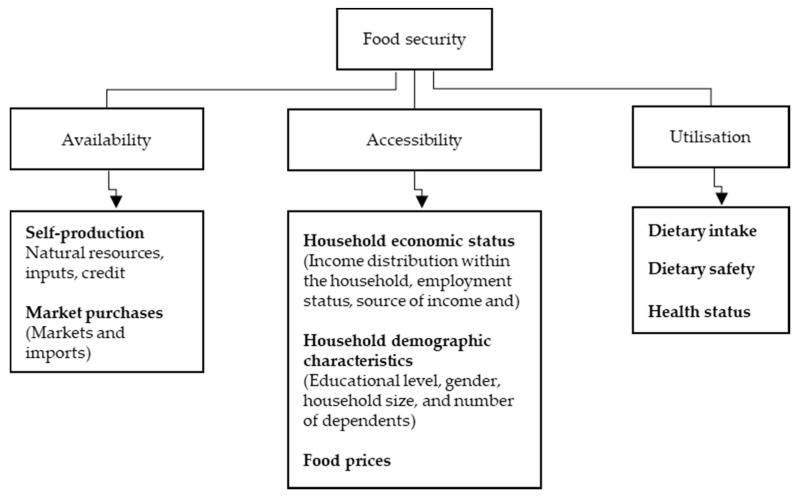
Conceptual framework of household food security. Source: Bashir and Schilizzi [18].

**Table 1 children-09-01521-t001:** Household with children and respondents (caregivers) demographic characteristics.

Age Distribution of Respondents (Caregivers)		
Age (years)	Frequency	Percentage (%)
<19	4	2.50
19–25	29	18.10
26–30	41	25.60
31–45	50	31.30
46–50	20	12.50
>50	16	10.00
Total	160	100
Respondents’ (caregivers’) highest level of education		
Level of education	Frequency	Percentage (%)
Never went to school	60	37.50
Non-formal education	7	4.40
Primary level	69	43.10
Secondary level	24	15.00
Total	160	100
Employment status of respondents (caregivers)		
Employment status	Frequency	Percentage (%)
Unemployed	111	69.40
Self-employed	21	13.10
Temporally employed	21	13.10
Permanently employed	7	4.40
Total	160	100
Respondent’s access to social grant		
Access to social grant	Frequency	Percentage (%)
Yes	153	95.60
No	7	4.40
Total	160	100
Household total number of dependent (s)		
Number of dependents	Frequency	Percentage (%)
1–5	84	52.50
6–10	76	47.50
Total	160	100

Source: Survey data (2019/20).

**Table 2 children-09-01521-t002:** The food insecurity access scale analysis results in households with children in the Inanda informal settlements.

HFIAS Category	Frequency	Percentage (%)
Food secure	12	7.50
Mildly food insecure	49	30.60
Moderately food insecure	44	27.50
Severely food insecure	55	34.40
Total	160	100
Mean	11.80	
Std. Deviation	7.52	

Source: Survey data (2019/20).

**Table 3 children-09-01521-t003:** Food groups consumed by households with children in the Inanda Informal settlements in a 24-h recall period.

Food Type Consumed by Households	Frequency	Percentage (%)
Cereal (bread, rice noodles, biscuits, or other foods made from millet, sorghum, maize, rice, wheat?)	159	99.38
Root and tuber (any potatoes, yams, manioc, cassava, or other foods made from roots or tubers?	143	89.38
Any vegetables?	99	61.88
Any fruits?	61	38.15
Meat, poultry, offal (beef, pork, lamb, goat, rabbit, wild-game, chicken, duck, or other birds, liver, kidney, heart, or other organ meats?	33	20.63
Any eggs?	40	25
Fish and seafood (any fresh or dried fish or shellfish?	19	11.88
Pulses, legumes/nuts (any foods made from beans, peas, lentils, or nuts?	89	55.63
Milk and dairy products (any cheese, yoghurt, milk or other milk products?	21	13.13
Oil/Fats (any foods made with oil, fat, or butter?	155	96.86
Any sugar or honey?	110	68.75
Miscellaneous (any other foods, such as condiments, coffee, or tea?	118	73.75

Source: Survey data (2019/20).

**Table 4 children-09-01521-t004:** Results of the household dietary diversity score of households with children in the Inanda Informal settlements.

HDDS Category	Frequency	Percentage
Low dietary diversity (<4)	123	76.90
Medium dietary diversity (4–6)	19	11.90
High dietary diversity (>6)	18	11.30
Total	160	100
Mean score	5.09	
Std. Deviation	1.28	

Source: Survey data (2019/20).

**Table 5 children-09-01521-t005:** Chi-square test and Pearson correlation between household food security status (HFIAS) and selected explanatory variables.

Variable	Food Secure *n* = 37	Mildly Food Insecure *n* = 41	Moderately Food Insecure *n* = 25	Severely Food Insecure *n* = 57	X^2^ Value	*p*-Value
Age of caregiver						
<19	2 (1.3%)	1 (0.6%)	0 (0.0%)	1 (0.6%)	61.878 *	0.000
19–25	4 (2.5%)	9 (5.6%)	11 (6.8%)	5 (3.1%)		
26–30	10 (6.2%)	15 (9.3%)	4 (2.5%)	12 (7.5%)		
31–45	17 (10.6%)	10 (6.2%)	4 (2.5%)	19 (11.8%)		
46–50	3 (1.8%)	5 (3.1%)	3 (1.8%)	9 (5.6%)		
>50	1 (0.6%)	1 (0.6%)	3 (1.8%)	11 (6.8%)		
Caregiver’s education level						
Never went to school	5 (3.1%)	12 (7.5%)	14 (8.8%)	29 (18.1%)	9.926 **	0.019
Non-formal education	2 (1.3%)	0 (0.0%)	2 (1.3%)	3 (1.8%)		
Primary	20 (12.5%)	20 (12.5%)	4 (2.5%)	25 (12.0%)		
Secondary	10 (6.2%)	9 (5.6%)	5 (3.1%)	0 (0.0%)		
Caregiver’s employment status						
Unemployed	18 (11.2%)	35 (21.8%)	20 (12.5%)	38 (23.7%)	42.090 *	0.000
Self-employed	9 (5.63%)	6 (3.75%)	0 (0.00%)	6 (3.75%)		
Temporally employed	7 (4.38%)	0 (0.00%)	4 (2.50%)	10 (6.25%)		
Permanently employed	3 (1.88%)	0 (0.00%)	1 (0.63%)	3 (1.88%)		
Caregiver’s access to social grant						
Yes	37 (23.1%)	39 (24.3%)	22 (13.7%)	55 (34.3%)	7.028 *	0.008
No	0 (0.00%)	2 (1.30%)	3 (1.88%)	2 (1.3%)		
Household total number of dependents						
1–5	22(13.7%)	19(11.8%)	10(6.2%)	33(20.6%)	1.679	0.950
6–10	15(9.3%)	22(13.7%)	15(9.3%)	24(15.0%)		

Source: Survey data (2019/20). * Significant at *p* < 0.1; ** Significant at *p* < 0.05; *n*—number of respondents; X^2^—Chi-Square value.

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
