# Peer review of "Food Insecurity in the Informal Settlements of Inanda Households Living with Children under 60 Months in Ethekwini Municipality"

_children, 2022, doi:10.3390/children9101521_

Round 1

Reviewer 1 Report

Review report

Title: Concise, simple, to the point, appeals to the reader and creates valuable information resource

Abstract: A well written summary of the study

Suggestion for consideration: Include summary of ethical consideration.

Keywords: Capture the essence of the study

Suggestion for consideration- Include Urban poor settlement, South Africa

 Introduction

Suggestions for inclusion

·         Extent of global, regional, and country specific food insecurity, associated factors

·         Food security and insecurity used interchangeably – definition of the two may help readers understand better

·         Study objectives/research questions would throw more light to the study

 Materials and methods (57 – 64):

Methodology: description of study area was comprehensive

Major area for consideration:

·         Relocate sections of methodology to introduction section (lines 68 -80, 78 - 107)

·         The study method described in confusing suggest rewriting. For example; mention of Quantitative cross sectional (line 109), but snow balling (line ?qualitative). Was this then both qualitative and quantitative methods

·         Sample size? deemed acceptable? (line 111) what sample size calculation was used?

·         IRB approval Not indicated yet important

·         Quality concert including pretesting of the questionnaire used

Results:

Key area for consideration

Suggest reorganization of the result area, as is, too difficult for a reader comprehend

 Discussion.

Area for consideration

·         It would be more interesting to compare findings with studies conducted in urban poor settlements, unless there is paucity of such studies.

·         Avoid repetition

·         Mention any limitation of the study

Conclusions: Too long, with embedded recommendations

·         Make conclusions, concise and simple to respond to the study objectives/questions

·         Separate Recommendation from Conclusions. Else, rename section as Conclusions Recommendations

Reference: Ok

Author Response

See the attached revision report.

Reviewer 2 Report

Generally original work.
However , lacks mainstream food security literature. Very much focus on specific region works.

Requires, language edition.

Method of measuring children food security was not appropriate.

The author took for grant where Ethekwini municipality is located across the world.

As the study considered households with under 60 months I expected anthropometric measurements for the children. Which is the most appropriate method?

Abstract - What does this integrated approach refer to?

The Inanda area was purposefully selected because of its poor
socioeconomic status, low agricultural potential, and
overpopulation. Bold statement is an indicator for high
level of food insecurity. It may question about justification of the study at the site.

Improve the quality of the map by showing where it is in RSA.

Author Response

See the attached revision report.
